# Evaluation of Gluten Exclusion for the Improvement of Rheumatoid Arthritis in Adults

**DOI:** 10.3390/nu14245396

**Published:** 2022-12-19

**Authors:** Avinent-Calpe Lidón, Martinez-López Patricia, Dhokia Vinesh, Massip-Salcedo Marta

**Affiliations:** 1Independent Researcher, 12003 Castellon, Spain; 2Techné Research Group, Department of Knowledge Engineering of the Faculty of Science, Universidad de Granada, 18071 Granada, Spain; 3Mechanisms of Ageing and Cancer Laboratory, Department of Molecular and Cell Biology, University of Leicester, Leicester LE1 7RH, UK; 4FoodLab, Faculty of Health Sciences, Universitat Oberta de Catalunya, 08035 Barcelona, Spain

**Keywords:** gluten, diet, rheumatoid arthritis

## Abstract

There is currently a growing anti-gluten trend which, except for individuals with coeliac disease and non-coeliac gluten sensitivity (NCGS) for whom its intake is contraindicated, results in gluten (the main protein in wheat and other cereals) being considered harmful to health and excluded from diets, largely due to information distributed through social networks. However, in many cases the recommendation to exclude gluten from the diet goes beyond personal choice and is promoted by health professionals. This choice and/or recommendation is especially important to individuals with chronic inflammatory diseases such as rheumatoid arthritis (RA), for which this exclusion is justified to reduce the symptoms of the disease. The aim of this literature review is to assess whether there is scientific evidence to justify the elimination of gluten in patients with RA, neither coeliac nor with NCGS, to improve their symptoms and quality of life. The results of the search on gluten and RA carried out in the Embase database and the extraction of data from 16 articles included in the review are presented. No scientific evidence was found to recommend the exclusion of gluten in patients with RA.

## 1. Introduction

### 1.1. Rheumatoid Arthritis

According to the Spanish Society of Rheumatology, the prevalence of rheumatoid arthritis (RA) in Spain is estimated at 0.5%, similar to other European countries, and three times more frequent in women [1]. It is estimated that RA affects around 5% of women over 55 years of age [2]. RA presents as a common form of arthritis that causes inflammation in the lining of the joints, resulting in warmth and redness, reduced range of motion, swelling, hypersensitivity and pain in the joints, and may cause damage to the cartilage, bones, tendons and ligaments of the joints [3]. Except for the spine (which is usually not directly affected by RA, except for the neck) and the distal phalanges, all other joints can be affected, and bilaterally.

The most common initial symptom is morning joint stiffness (especially in the joints of the hands and feet), which occurs after a night’s rest and leads to significant difficulty in movement. It is therefore a chronic inflammatory disease, autoimmune in nature, characterized by symmetrical involvement of multiple joints and the presentation of various non-specific general symptoms and extra-articular manifestations (skin, blood vessels, heart, lungs, eyes, blood). Left to its natural course and the absence of adequate treatment, the disease can cause, in advanced stages, significant physical limitations as well as a marked impairment in quality of life [2].

Although the causes of RA are not fully understood, it is known that genetic factors (polygenic disease) and non-genetic components (infections, hormones, smoking, stress, obesity, diet) are involved in its origin [2]. In RA, it is known that the body’s immune system (IS) plays an important role in inflammation with the IS itself attacking the joints, through the invasion of immune cells causing inflammation of the synovial tissue [3]. It can therefore be described as a systemic autoimmune disease, as the lesions it causes involve the body’s connective tissue, a tissue that is present in virtually all organic structures [2]. The production of various enzymes, antibodies, cytokines, etc. will attack all joint structures leading to their deformation.

Effective treatments exist for both symptom relief and to modify the course of the disease; pharmacological therapy usually involves a combination of drugs (analgesics, anti-inflammatory NSAIDs and corticosteroids and DMARDs) for prolonged periods of time [2]. As these drugs may affect the IS or have pronounced side effects, careful medical supervision during treatment is necessary. Algorithms to aid decision-making in health care, without being mandatory or replacing the clinical judgment of the professional involved, are outlined in the Clinical Practice Guideline for the Management of Patients with Rheumatoid Arthritis [4], where the SIGN levels of evidence and grades of recommendation are also very well reflected.

In addition to medication, and as part of non-pharmacological therapy, it is important that those affected take regular moderate physical exercise to prevent loss of joint mobility, reduce fatigue, strengthen muscles and bones, increase flexibility and improve overall well-being. The multidisciplinary approach required for RA includes, amongst others, dietary measures [1], which not only involves following a healthy diet but also the intake of certain nutrients as will be discussed later. Additional general recommendations for patients who do not yet have joint damage are similar to those given for the general population: not smoking, sleeping a minimum of 8 h, moderation in alcohol consumption and avoiding activities with intense or sustained physical exertion [2].

### 1.2. Arthritis and Diet

The role of diet in preventing and treating different types of arthritis is not yet known, although diet will clearly be a key factor to consider in patients, for example, who are overweight and/or those with low calcium intake and/or those with excessive protein intake, etc. Similarly, there are indications that certain dietary components may reduce inflammation, such as omega-3 acids, or limit the progression of osteoarthritis, such as diets high in vitamin D and C [3]. A Mediterranean diet and consumption of olive oil are of great interest due to their high antioxidant and anti-inflammatory capacities [1]. However, in some instances the evidence is not clear, as reflected in the meta-analysis by Genel et al. [5]; the authors analyzed the impact of anti-inflammatory dietary interventions, based on the principles of a Mediterranean diet, concluding that although the anti-inflammatory diet is associated with increased weight loss, decreased inflammatory biomarkers, improved joint pain and improved physical function, the evidence is really poor, recommending the need for high quality studies led by nutritionists.

Food allergies can also trigger symptoms of arthritis [3]. Since an allergy is an adverse reaction of the immune system to the ingestion of a food (by contact with the allergens, proteins, contained in food), this would increase general inflammation and be detrimental to arthritis. This may explain a trend in naturopathy courses and by some authors [3,6] to promote vegan diets, which recommend the elimination of foods considered in these forums as pro-inflammatory (aubergines, tomatoes, potatoes, peppers), and recommending a healthy diet based on fruits, vegetables, whole grains, nuts and seeds, with a higher intake of broccoli, Brussels sprouts, cabbage, garlic, turmeric, sour cherries and vitamin C, foods that help fight inflammation and joint pain. While it is generally recommended to eliminate refined grains, which are associated with higher levels of inflammatory markers in the blood, and replace them with whole grains (both in their direct consumption and derivatives), research has not confirmed any connection between whole grains and inflammation; however, there are many other good reasons for their consumption. Although some RA patients may show some improvement with the elimination of some foods, there is not enough evidence to recommend specific diets [7].

### 1.3. Arthritis and Gluten

Books such as “No grain, no pain” [8] or “Grain Brain: The surprising Truth about Wheat, Carbs, and Sugar—Your Brain’s Silent Killer” [9], in which the authors, who are not nutritionists, claim that cereals should be totally excluded from the diet have been published. Their dissemination through social networks has generated in some sectors an aversion to the intake of carbohydrates, the main nutrient in most cereals, leading to ketogenic diets amongst others to entirely remove cereals. It is worth noting the risk involved in this type of diet without consulting a specialist, as few carbohydrates are consumed, which, nutritionally speaking, is a mistake. Cereals also provide proteins, a low percentage of lipids, vitamins, and minerals. Of the proteins, we should highlight gluten, a non-soluble protein, which, like other proteins, can trigger allergic or inflammatory reactions in sensitive patients.

The increase in cases of food intolerances, one of them being gluten [10], has made this protein fashionable, or rather being non-gluten fashionable. Going from a term only known to coeliacs to whose removal is indicated even in foods that are naturally gluten-free; from being an unknown term to being the hypothetical cause of a multitude of inflammatory processes and other pathologies, thus becoming the new villain of the diet. So much so, that nowadays it is not only people with coeliac disease who buy gluten-free foods, but also the whole family and consumers who have not been diagnosed or have an intolerance but have opted to consume gluten-free products, leading to a growing demand for foods made with gluten-free flours.

Could the food industry be exploiting and taking advantage of this *gluten-phobia*? The number of available gluten-free foods have increased enormously: in the USA, sales of gluten-free products have increased almost sixfold in a decade as substitutes are significantly more expensive [11]. In the UK, 60% of adults have bought a gluten-free product and 10% of households have a member who thinks that gluten is bad for their health [12]. From a need for coeliacs, it has become a consumer trend. As López Iturriaga pointed out in 2013 [11]: “Poor gluten. Without eating or drinking it, […] today many feel it as a threat to their health […] when in reality it is only harmful for those with allergies and coeliac disease”. Levinovitz, in his book “*The gluten lie and other food myths*” [13] talks about the nocebo effect for gluten.

Some dietary portals state that gluten should be avoided to improve arthritis [14] as a gluten-free diet could reduce the accumulation of joint fluid. Another example would be the claim that gluten causes inflammation in the joints of people with RA and worsens symptoms, recommending the selection of gluten-free foods [15]. Although some studies correlate gluten elimination with a reduction in antibody titre [16], which may be expected, no difference in cartilage destruction rates is observed, which would indicate that gluten elimination has no effect. However, social networks only show the first part of the results [17] and, although they reference the article (as proof of their claim), who follows up and reads the original article? Similarly, although reference is made to the fact that a decrease in intestinal permeability is beneficial for RA patients and that wheat increases intestinal permeability, due to its component in gliadins [18] and lectins [19], the studies are laboratory models and not clinical trials in humans, which means that these results cannot be extrapolated to real patients. This cannot be surmised upon reading of the article on social networks, leaving amateur readers with a notion of the harm caused by wheat, in arthritis patients. This growing anti-gluten trend is not only found in the general public (who may be more likely to follow fashionable trends), but also professionals in the nutrition sector (via recommendations) and by many people who practice naturopathy (whether they are health professionals or not).

Therefore, in this paper we want to provide the existing evidence on what the exclusion and inclusion of gluten in the diet would cause in adults with RA who do not have coeliac disease and who are recognized not to be sensitive to gluten, in order to clarify, as will be seen in the following section, whether exclusion of gluten is beneficial for people with rheumatoid arthritis.

## 2. Analysis and Methodology

The relationship between gluten and RA was carried out by means of a systematized process [20] based on the search for scientific articles in specialized electronic databases, in a sequential process [21], having previously defined the specific search objective [22,23]. PRISMA guidelines [24] were followed for the writing of this literature review. To conclude the studies analyzed, a blinded peer review was carried out with the Abstrakr tool in beta function.

### 2.1. Information Sources

The bibliography used in the review was extracted from the Embase database in December 2021. Initially, PubMed, Scopus and Epistemonikos databases were also consulted, but these were discarded as Embase was considered to contain the same results.

### 2.2. Search Strategy

The search strategy was carried out by including articles in Spanish and English, with no time limit with articles between 1991 and 2021, which included in their title or abstract the words “rheumatoid arthritis” AND “gluten”, with the search key being “(‘rheumatoid arthritis’/exp OR ‘rheumatoid arthritis’) AND (‘gluten’/exp OR gluten)”.

### 2.3. Selection of Studies

The *Abstrackr* program in its Beta version was used to initially screen articles according to their “*abstract*”. This selection was carried out independently by 2 reviewers who, based on the eligibility criteria (see Table 1), confirmed whether or not the article met the inclusion criteria. To facilitate an objective selection, a table of inclusion and exclusion criteria was created. Discrepancies were reviewed where they existed and accepted/rejected at the author’s discretion.

Finally, articles that met all requirements were included in the final write-up of the results and discussion. The AMSTAR critical appraisal tool [25,26] was used for the systematic review included in this paper.

## 3. Results

Figure 1 shows the process in the selection of articles for this review. Of the 288 articles located in Embase, 257 were excluded during the Abstract review because they did not meet the exclusion criteria. A total of 31 articles were therefore selected for full-text review and data extraction; during the reading process, 5 articles were found which, due to references in the text and title, were included in the full-text review group. Out of the total of 36 articles selected, 19 were not considered, 7 because they did not meet the inclusion criteria and 12 because they could not be found. Thus, 16 articles were finally selected for data extraction and preparation of this literature review. Table 2 shows the main characteristics of the included studies, with the variables analysed in each one (if applicable), articles that have been published from 1991 to 2021.

In 1991, Kjeldsen-Kragh et al. [27] published the first article to which we have had access and discusses the gluten-free diet-rheumatoid arthritis (GFD-RAR) binomial. In a 13-month prospective study, 53 patients were divided into two groups, one undergoing a GFD (27 people) and the other (26 people) acting as a control group. The NGD starts with a 7–10 day fast (daily energy intake 800–1260 kJ), whilst introducing a new food every 2 days; during the first 3–5 months no gluten-containing foods (no meat, fish, eggs, dairy, refined sugars or citrus fruits) were allowed. Patients in the control group took an ordinary (unspecified) mixed diet. The main results according to the authors, for the 34 patients at the end of 13 months, were significant improvements in the intervention group for all variables (except platelets and haemoglobin); however, in both groups there was radiographic deterioration at the end of the period and no differences between the groups. Greater weight loss, lower haemoglobin values and nutritional deficiencies (until the introduction of the lacto-vegetarian diet) were observed in the intervention group.

The same group of researchers published an article in 1993 on the influence of fasting and a vegetarian diet (with a gluten-free period) on the nutritional status of RA patients [28]. The patient group is the same as their previous work [27]. It should be noted that in this work the main comparisons are intra-group rather than inter-group, especially the anthropometric variables for which there is no inter-group comparison, with particular reference to the values before and after intervention for each of them. For analytical variables, no differences were detected between the groups. A low energy and protein intake was detected in the intervention group, with a decrease in anthropometric measurements. An assessment of nutritional status is therefore recommended as mandatory before starting fasting and diet manipulations and requires medical supervision and dietary support. Patients found it difficult to follow the gluten-free vegan period.

In the same clinical trial, but in a different publication [29], the mechanisms that could explain the effect of the vegetarian diet were analyzed. They do not rule out the existence of a possible placebo effect since some of those who responded positively to the vegetarian diet believed less in the effect of ordinary medical treatments. On the other hand, the results, presented and published in detail in several previous articles by the author and his collaborators, do not indicate an effect of food antigens, with no changes in fatty acid concentrations found between those who responded positively to the vegetarian diet and those who did not. The possible stimulation of the immune system by antigens of the gut commensal *Proteus mirabilis* were also summarized. For the vegetarian diet this appears to decrease IgG activity. Changes in fecal flora were also observed and hypothesized that this could lead to differences in the absorption of substances which in turn influenced inflammatory processes in the joints.

Hafström et al. [16], in a single-blind randomized clinical trial, studied the effect of a gluten-free vegan diet (GFDV) compared to a non-vegan diet, with foods from all food groups, to test for improvement in RA signs and symptoms. The 12-month trial did not show that the vegan diet exerted a joint protective effect, although given the small number of patients the power of this analysis is rather limited. It is concluded, on the other hand, that the effects on arthritis correlate with a reduction of antibodies to food antigens (anti-gliadin IgG and β-lactoglobulin) in patients responding to the VVSDG, although the authors mention that they cannot exclude that the improvement does not have a causal connection with the decrease of antibodies; in fact, the mere existence of food antibodies does not indicate disease.

As RA patients have a higher cardiovascular risk, Elkan et al. [30] investigated the effects of a gluten-free vegan diet on LDL-cholesterol and natural atheroprotective antibodies against phosphorylcholine (anti-PCs). The results showed that patients in the vegan diet group had decreased BMI, weight and LDL-cholesterol levels, positive factors for a lower incidence of CVD. However, the change in lipid profile was a consequence of the vegan diet and not the result of reduced inflammation. Oxidized LDL (oxLDL) values tended to be lower in the vegan group and, given that oxLDL has pro-inflammatory properties, its reduction may have contributed to the decrease in disease activity in this group. Furthermore, oxLDL is a risk marker for CVD and atherosclerosis, and its reduction contributed to a less atherogenic profile as did the lower anti-phosphorylcholine (anti-phosphorylcholine) values.

The work of Lidén et al. [31] contained a double study with self-perception surveys in patients with RA and self-reporting of food intolerance in addition to a rectal test of sensitivity to milk and gluten being performed, comparing the effect in patients with RA and healthy patients. The results show, for the first part of the study, that 9% of respondents report intolerance to cow’s milk and meat (the two foods reported as most intolerant in the survey) and 5% to wheat. However, the authors admit a bias may exist due to these foods in popular culture and in the medical literature, having been reported as having an unfavorable influence on RA symptoms. For rectal exposure, no differences in antigen levels were detected; the authors put the possibility down to the low sensitivity and specificity of the tests used, and comment that the more specific tests are not often used in clinical practice. They admit the existence of confounding factors, as patients were not classified by age or sex and were treated with drugs (decreased intestinal mucosal reactivity). Mucosal sensitivity to gliadin was observed only in a minority group of RA patients. Serum antibody analysis did not seem to prove an immunological link between diet and RA as activation of the intestinal immune system is not reliably reflected in serum.

The Cochrane systematic review by Hagen et al. in 2010 [32] analyses 15 studies by other researchers in which fasting, vegetarian (usually gluten-free) diets, the Mediterranean diet, or elemental liquid-based diets or elimination diets are used to improve RA symptoms. Among the studies included in the review are some of those already discussed previously, such as Kjeldsen-Kragh et al. 1991 [27] and Hafström et al. 2001 [16], both with a moderate risk of bias as indicated in this review. The results of the review reflect that it is uncertain whether diets improve pain, stiffness, and the ability to move better, as the studies included in the review are small trials with a moderate risk of bias. Conversely, diets can be difficult to follow and weight loss can occur even without planning, which can put patients at nutritional risk. The is a need for more robust research on dietary interventions for RA. More powerful studies need to be designed with long-term follow-up, including possible adverse effects. Although the elimination diet is included in this review, there is no review analyzing the effect of gluten elimination per se. The application of the AMSTAR-2 tool resulted in a “Low” confidence level for this systematic review, as one critical weakness (item 2) and some non-critical weaknesses were found.

El-Chammas et al. [33] review gluten-free diets in non-coeliac diseases, including RA. The section on this disease refers to the immunological activity of some antibodies and the CV risk and atherosclerosis of these patients, based on the work of Hafström et al. [16] and Elkan et al. [30], briefly commenting on the variables studied in them. They conclude that, for RA, the real role played by gluten per se is unknown.

Several factors prompted Lerner et al. [34] to review the literature on the adverse effects of gluten and the advantages of gluten elimination in non-coeliac autoimmune diseases, including RA. These included the importance of wheat in human nutrition, especially in Western diets, that gluten is also a major food additive in the processed food industry, that the prevalence of gluten-related disorders has been increasing over the last 30 years and that there is some evidence that gluten may be related to the increasing incidence of autoimmune diseases other than coeliac disease. The authors comment that, given that there are many studies describing the similarities between coeliac disease and RA, it is logical that the elimination of gluten would also be beneficial in patients with arthritis. They refer to the work of Hafström et al. [16] and Elkan et al. [30] as studies demonstrating reduced immunoreactivity to food antigens and that a gluten-free diet is potentially atheroprotective and anti-inflammatory in patients with RA. However, one should not forget the confounding factors that may occur in the aforementioned articles, as the diet used is not excluding gluten alone but is the consumption of a vegan diet and the change in lipid profile observed was a consequence of a vegan diet and not of reduced inflammation. The authors themselves conclude that there is currently no clear indication for eliminating gluten in non-gluten related conditions as the evidence for the detrimental effects of gluten is limited to a few in vitro and in vivo studies and that more controlled studies in humans are needed.

Badsha’s article [35] reviews the literature on the impact of different diets and dietary interventions on RA (Mediterranean diet, flavonoids/isoflavones, gluten, elemental diet, vegan, elimination, polyunsaturated fatty acids, probiotics, alcohol, vitamins, supplements, antioxidants, fasting, obesity). Their conclusion is clear: dietary interventions for RA are limited and not of high quality; most are heterogeneous in intervention and variables to be studied, as well as unbalanced at baseline and with inadequately reported data [32]. Limited evidence to associate gluten-free diets with benefits in RA.

The review by Vijayalakshmi [36] simply refers, for a gluten-free diet, to the work of Hafström et al. [16] commenting only that the gluten-free vegetarian diet could be beneficial for some patients as indicated in the article referenced by Vijayalakshmi.

Lerner et al. [37] revisit the question of whether gluten-free diets are worthwhile in patients without coeliac disease. To do so, they review evidence for these diets and their efficacy in sports performance, neuropsychiatric disorders and RA; they also review the potential disadvantages of adhering to such gluten-free diets. For RA, they refer, among others, to the systematic review by Hagen et al. [28] and the work of Kjeldsen-Kragh et al. [27], Hafström et al. [16] and Elkan et al. Furthermore, like other authors, they report that despite its popularity, the gluten-free diet has not been studied in isolation in patients with RA and is premature to conclude that it is beneficial in this context. In the current evidence, the effect of gluten exclusion is confounded with the exclusion of products such as meat and dairy. For their part, they are keen to point out that several studies have shown that such diets are often deficient in whole grains, fibre, and micronutrients and may even contain more sugars and saturated fats than their gluten-containing counterparts. In fact, while gluten intake by itself is not associated with risk of coronary heart disease, avoiding gluten in the diet can lead to deficiency of whole grain intake which in turn has cardiovascular benefits. However, as people who do not eat gluten-containing grains eat more rice and fish, urinary metal residues are often higher and, additionally, these people may develop anxiety.

Bruzzese et al. [38] report four cases of patients with long-term RA in whom biological medication was not producing the expected effect and whose diet was changed to a gluten-free one to see if there was any improvement in their condition (DAS28). Although all four cases reported an improvement (see table), this can only be followed for up to 10 months in case 1, as in cases 2 and 3 the patients abandon the diet and in case 4 there is no mention of what happens after 10 days. According to the authors, the gluten-free diet cannot be considered sufficient therapy for inflammatory autoimmune diseases but can serve as a complementary support. Correct answers on nutrition in these diseases are not yet available.

The literature review by Rondanelli et al. [39] evaluates the evidence up to 2020 regarding the ideal diet for the management of RA to reduce associated inflammation and to construct a food pyramid for these patients. It includes 227 articles in its review, with varied types of studies, whose level of evidence it also evaluates. The results and discussion are grouped by types of food, giving the evidence for or against each group; although there is no specific section for gluten, carbohydrate intake is discussed (review of 12 articles). Their discussion focuses on the similarities between RA and coeliac disease and how intestinal inflammation goes beyond the intestine itself (review of several immunology articles), on carbohydrate intake, especially potatoes and sugars in beverages, and on cardiovascular risk factors in this type of patient. In their conclusion and recommendation for carbohydrate intake they suggest the exclusion of sugar and sweeteners from the diet and consume three portions per day of whole grain cereals, preferably gluten-free; however, no evidence is provided for the latter recommendation.

Guagnano et al. [40] conducted a clinical trial in well-controlled, medicated RA patients, in which one group excluded meat, gluten and dairy products from their diet and the other did not; the patients, 40 in total, usually consumed a Mediterranean diet. The aim was to analyze whether this exclusion improved the health of patients, their joints, as well as several analytical, anthropometric and immunological parameters (Table 3 shows the variables analyzed). The results showed some intra-group differences: for the exclusion diet group there are differences in VAS and HAQ pre-post diet indices; there are no differences in DAS28. In anthropometric measures, inter-group differences are only observed for muscle mass, water and systolic pressure. For blood tests there are no inter-group differences. For cytokines and adipokines, with no inter-group comparison, there is variation only for leptin in each of the two groups. The study encountered several limitations given the low number of patients completing the trial [16] and that all of them were overweight or obese women, so they may not represent all cases; furthermore, the intervention period was short. Once again, we are again faced with confounding effects for an exclusion diet, given that the differences observed cannot be attributed to any of the excluded foods in particular and that the exclusion of gluten is responsible for them.

Jiang et al. [41] conducted a literature review to assess the effect that different diets (Mediterranean, vegetarian and vegan, gluten-free, fasting, specific carbohydrate, low FODMAP, paleo, Atkins) could have on irritable bowel syndrome but extended to other diseases, such as RA. In the case of a gluten-free diet and its effect on RA, they refer to the work of Kjeldsen-Kragh et al. [27] and Hafström et al. [16] as studies that report improvement in RA symptoms, indicating that there are no published studies with diets that exclude only gluten. They also comment that while gluten-free vegetarian diets appear to be beneficial for RA patients, these studies include potential confounding factors.

## 4. Discussion

The results of this literature review show that much, if not all, of the information available on the effects of gluten exclusion in non-coeliac rheumatoid arthritis patients without NCGS is based on studies in which, although gluten has been eliminated from the diet, this has not been the only variable modified, so that confounding bias was present in all the literature found, the exception being the study of four cases by Bruzzese et al. [38], which, due to a multitude of factors (very small number of patients, design, short duration) cannot be considered representative. In five clinical trials [16,27,28,29,30] the intervention is gluten-free but within a vegan diet, which is compared with other diets (not always well defined). For rectal mucosa exposure [31] there is simultaneous exposure to milk proteins and gluten, and in the work of Guagnano et al. [40] there is joint exclusion of red meat, gluten and dairy products. In all cases, the trials are also small in population sample size and with moderate to high risk of bias [42]. However, the systematic review [32] and the literature reviews [33,34,35,36,37,39,41] can only mention these studies, and therefore most refer to the work carried out by Kjeldsen-Kragh [27,28,29], Hafström et al. [16] and Elkan et al. [30]. In their conclusions, the authors of these reviews indicate that:- The actual role of gluten per se is unknown,- That there is no clear indication for eliminating gluten in non-coeliac patients,- That the evidence is limited for associating gluten-free diets with benefits in RA,- As the gluten-free diet has not been investigated in isolation it is premature to conclude that it has benefits for RA.

This issue of confounding is also reflected in the work of Aho et al. [43] whose authors, in a review article on risk factors for RA, comment on the impossibility of distinguishing the effects of various nutrients from each other or separating the effect of diet from lifestyle factor rendering the observed diet-RA associations invalid. It is certain that nutrition affects RA, but identification of crucial nutrients may not be possible through epidemiological studies.

So why is it assumed that gluten exclusion would be beneficial for RA patients? Possibly because it is identified with improvements in coeliac disease, for which gluten is a trigger and for which gluten exclusion is a lifelong treatment (as it would be for NCGS). Both RA and coeliac disease belong to the family of autoimmune diseases and share epidemiological aspects [44]. For example, incidence and geo-epidemiological trends are similar, females being more affected (even more so in RA), the importance of stress in the clinical manifestations, triggering by infections, and both being associated with other autoimmune diseases (multiple sclerosis, Sjogren’s syndrome, type 1 diabetes, systemic lupus erythematosus, etc.). Additionally, family history is an important risk factor in both (although in both cases the genetic predisposition is related to haplotypes of the human leukocyte antigen system, HLA (*Human Leukocyte Antigen)*, for RA it would be the DRB1 haplotype while for coeliac disease the related haplotypes are DQ2 and DQ8). Intestinal inflammation is another common feature and there may be inflammation beyond the initial tissue (intestine in coeliac disease, joints in RA). In both diseases there are changes in the enteric microbiome, with a decrease in diversity (especially of *Bifidobacterium* and *Lactobacillus*) though populations are different. According to the work of Lerner et al. [44] there are multiple observations suggesting that RA is an inflammatory state that starts in the gut, beginning years before joint manifestations are detected. However, these same authors, as a therapy for RA and coeliac disease, exclude gluten from the diet of coeliacs but not of RA patients.

However, some patients with coeliac disease have symptoms typical of RA, which diminish if gluten is withdrawn [45]; though as these patients are coeliacs, adds little to our review. This same paper refers to the following statement from another paper by their group [34]: “...the gluten diet has recently been described as pro-inflammatory, pro-oxidative, anti-apoptotic and even activating the innate immune system with negative effects on T regulatory cells”, though referring to coeliacs, an aspect overlooked when non-professional media extract the information. If we combine this with a heading of the same work [34] entitled “Adverse effects of gluten on human health” we already have a headline. The hypotheses (not evidence) assume that since coeliac and many other autoimmune diseases share HLA genes, it is conceivable that this genetic load transfers some gluten toxicity to other autoimmune diseases and together with the other characteristics described [44], it is interesting to investigate the therapeutic effect of gluten-free diets in non-coeliac patients with autoimmune conditions. The conclusions again show that there are no clear indications for the elimination of gluten in these patients; what is not clear is that the disseminators fully read the document.

And while it is true that gluten sensitivity may be more common in patients with RA than in the general population [46], this does not mean that all patients suffer from it; clearly, for patients with this sensitivity, gluten exclusion would be necessary. The evidence for the NCGS-RA association is weak, as the cases reported in the work of Losurdo et al. [47] represent only a low level of evidence and case-control studies are needed.

In the present study, we have found a similar situation to that indicated by San Mauro et al. [48] in their review on the effect of gluten on neuronal diseases: *“an immense number of inconsistent results and little scientific evidence have been found that is causing the general population to choose gluten-free products as a healthier dietary pattern, although we have not been able to find this association in the present review”,* although in our case the information available to date is very scarce. Similarly, Palmieri et al. [49] conclude that a gluten-free diet could provide benefits in patients with NCGS and irritable bowel syndrome, but there is no solid evidence to indicate this diet in endocrinological, psychiatric, rheumatological diseases or to improve sports performance.

However, we must not forget the nutritional implications of a gluten-free diet, as the foods eaten tend to be richer in fat, sugar and salt than their gluten-free counterparts, as well as being a lower source of protein and fibre. Additionally, the amounts of some minerals and vitamins (Fe, Ca Se, Mg, Zn, folates nicacin, thiamine, riboflavin and vitamin A and D) have been found to be inadequate in patients who follow this diet without professional advice [50]. Furthermore, fructan-type resistant starches (oligofructose and inulin) support the maintenance of a balanced and healthy microbiome, which in turn protects against inflammatory conditions, among others [51]. It is also important to consider the economic impact, as these products are priced up to 500% higher than their gluten-containing counterparts [50].

## 5. Conclusions

No scientific evidence has been found to promote the exclusion of gluten in rheumatoid arthritis patients without coeliac disease or NCGS. Clinical trials with exclusion diets, with gluten as the only factor to be excluded, are needed to demonstrate conclusively whether gluten exclusion improves the symptoms of patients with rheumatoid arthritis.

## Figures and Tables

**Figure 1 nutrients-14-05396-f001:**
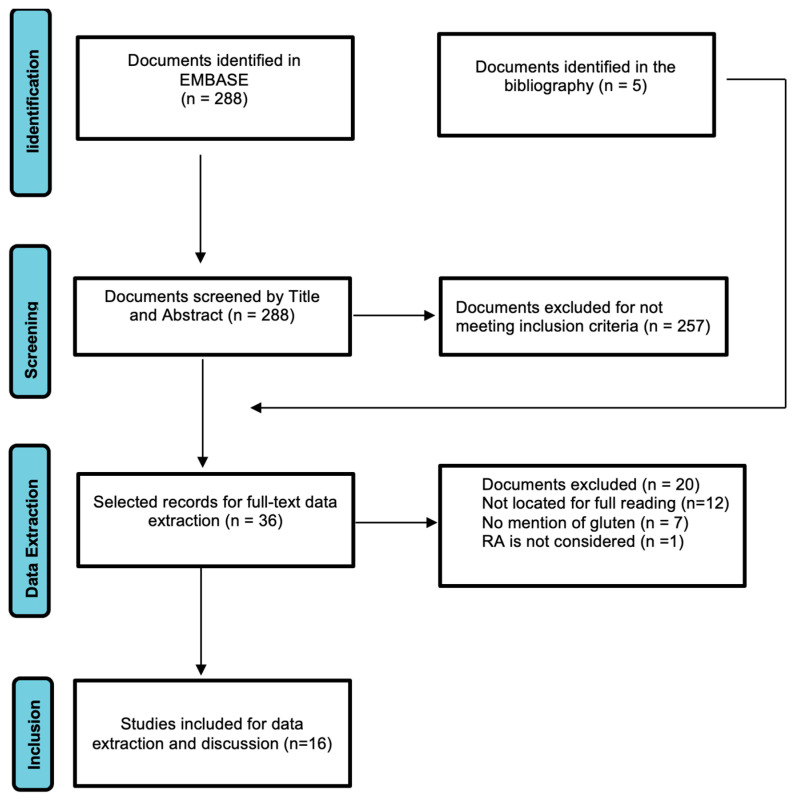
PRISMA Flowchart of screening and section of articles.

**Table 1 nutrients-14-05396-t001:** Eligibility criteria.

Population	Adults with Diagnosed Rheumatoid Arthritis
Intervention/Exposure factor	Exclusion of gluten from the diet for the improvement of symptomatology.
Comparator	Adults with gluten-free diets.
Patient-centered “Outcomes” of interest	Observation of improvements in arthritis symptomatology experienced by patients following dietary changes.
Type of design	Systematic reviews, meta-analyses, randomised controlled trials, randomised controlled trials, cohort studies, case-control studies, case-control studies

**Table 2 nutrients-14-05396-t002:** Inclusion and exclusion criteria.

Evaluation of Gluten Exclusion for the Improvement of Rheumatoid Arthritis in Adults						
**ELIGIBILITY CRITERIA**: screening according to information available in title and/or abstract and study design						
Is the elimination of gluten from the diet a factor of improvement in the painful symptomatology of rheumatoid arthritis in adults versus non-elimination?						
What is the relationship between gluten consumption and the development of rheumatoid arthritis?						
Is non-coeliac gluten sensitivity associated with the development or progression of rheumatoid arthritis?						
**INFORMATION**					**YES**	**NO**
Specifically rheumatoid arthritis and gluten are discussed.					√	
Other arthritis is considered but not rheumatoid arthritis.						√
Other arthritis in addition to rheumatoid arthritis is considered to include					√	
Diets in general are discussed without specifying gluten.						√
Coeliac disease and/or gluten sensitivity is treated but not arthritis.						√
Gluten sensitivity and arthritis treated					√	
Patients are not adults						√
**DESIGN**					**YES**	**NO**
Human studies						
Meta-analysis					√	
Systematic Review					√	
Observational Studies				Report and case series		√
				Transversal		√
				Population		√
				Cases and controls	√	
				Cohorts	√	
Experimental Trials				Randomised, blinded	√	
				Randomised, double-blind	√	
				Non-randomised		√
Animal studies (laboratory)						√
“In vitro” studies						√
The Summary will be ACCEPTED if: there is a YES in the information section and a YES or doubt a priori (?) in the design section, or if we doubt (?) the information but the design has a YES or? or if there is only a title (no Abstract) and we don’t know if it is of interest or not (?) In all other cases the Summary will not pass the screening.						
	**Information**					
	SI	NO	REVIEW			
**Design**						
SI	ACCEPT	REJECT	ACCEPT			
NO	REJECT		REJECT			
REVIEW	ACCEPT		ACCEPT			

**Table 3 nutrients-14-05396-t003:** Main characteristics of the studies/articles included in the review included in the review.

Author Year	Population (*n*) Duration * Type of Study	Intervention * vs. Control	Diet	Variables	Results
Kjeldsen-Kragh et al. 1991 [27]	*n* = 53 13 months ECASC	Fasting (7–10 days) + DVSG (3–5 months) + DV Vs. DMO	A: herbal teas, garlic, vegetable broths, potato and parsley decoction, carrot juice, beetroot, celery. DVSG: A + introduction of gluten-free plant-based foods. DV: DVSG + dairy and gluten free DMO: not specified	Pain, duration morning stiffness, personal feeling of condition vs. initial condition, no. painful joints, swollen, grip strength, Hb, platelets, ESR, white blood cells, CRP, serum albumin	A total of 34 patients finished Significant improvements in IG for all variables except platelets and Hb. Radiographic deterioration occurred in both groups with no differences between them. >weight loss, ↓Hb and some nutritional deficiencies in GI
Hagen et al. 1993 [28]	*n* = 53 13 months ECASC	Fasting (7–10 days) + DVSG (3–5 months) + DV Vs. DMO	A: herbal teas, garlic, vegetable broths, potato and parsley decoction, carrot juice, beetroot, celery. DVSG: A+introduction of gluten-free plant-based foods. DV: DVSG+ dairy and gluten free DMO: not specified	Weight, height, upper arm circumference, PCT, BMI, Hb, albumin, Cu, Zn, ferritin, IGF1, tryptophan	For anthropometric measurements, the two groups are not compared. For analytical variables no differences were detected between the groups. A low energy and protein intake was detected in the intervention group, with a decrease in anthropometric measurements. Patients found it difficult to follow the gluten-free vegan period.
Kjeldsen-Kragh 1999 [29]	*n* = 53 13 months ECASC	Fasting (7–10 days) + DVSG (3–5 months) + DV Vs. DMO	A: herbal teas, garlic, vegetable broths, potato and parsley decoction, carrot juice, beetroot, celery. DVSG: A + introduction of gluten-free plant-based foods. DV: DVSG + dairy and gluten free DMO: not specified	Pain, duration of morning stiffness, personal feeling of condition vs. baseline, no. painful joints, swollen, grip strength, weight, Hb, platelets, ESR, white blood cells, CRP, serum albumin, fatty ac., IgG *P. mirabilis*, faecal flora	Possible placebo effect on DV. The data do not indicate that systemic immune reactions against food antigens were significant in most patients. The clinical effects of VD do not appear to be due to changes in eicosanoid precursors. Possible stimulation of the immune system by *Proteus* antigens. Changes in faecal flora possibly leading to differences in the absorption of substances which in turn influence inflammatory processes in the joints.
Hafström et al. 2001 [16]	*n* = 66 12 months ECASC	DVSG (*n* = 38) Vs. DE (*n* = 28)	DVSG: Vegetables, dried and fresh fruits and nuts, maize, rice, sunflower seeds, buckwheat, millet DE: variety of food from all food groups	Patient status ACR Food antibodies (IgG, IgA against gliadin and β-lactoglobulin) X-rays	According to the ACR, in the DVSGs there is improvement in all indicators except CRP; in the EDs, improvement is observed only in joint swelling and global physical condition. Decreases Ig in DVSG (but only in dietary responders) No difference in radiological progression
Elkan et al. 2008 [30]	*n* = 66 12 months ECASC	DVSG (*n* = 38) Vs. DE (*n* = 28)	DVSG: Vegetables, dried and fresh fruits and nuts, maize, rice, sunflower seeds, buckwheat, millet DE: variety of food from all food groups	BMI, DAS28, physical fitness, ESR, CRP, Hb, WBC, serum albumin, total cholesterol, LDL, HDL, TG, OxLDL, anti-CCP	BMI, weight, LDL < at DVSG Anti-PC Ig M > DVSG Increases Anti-PC Ig A ↓ OxLDL (trend) DVSG DAS28 > DE
Lidén et al. 2010 [31]	*n* = 241 EP	Perceived personal connection between food and different symptoms	---	Intestinal, urticaria, itching, eczema, dyspnoea, rhinitis, anaphylaxis, fatigue, joint and muscle symptoms, etc.	Overall, 27% of patients reported food intolerances, mainly to cow’s milk (34%) and meat (33%); wheat ranked 5th (17–5% of all respondents).
	*n* = 45 CCT	AR (*n* = 27) Vs. no AR (*n* = 18)	Rectal exposure to milk and gluten	MPO, ECP, NO IgA + IgG antibodies against casein, β-lactoglobulin, α-lactoalbumin, gliadin, transglutaminase	No increase in MPO, DBS, NO after the intervention. Similar antibody levels in the two groups. Mucosal sensitivity to gliadin only seen in a minority group of RA patients.
Hagen et al. 2010 [32]	RS *n* = 837 15 studies (RCT–CCT)	---	A, DMO, DVSG, DV, MD, ED, eD	Pain, functional status, joint stiffness, fatigue, weight loss, gastrointestinal symptoms, sick leave, quality of life	There is no certainty that diets improve pain, stiffness and the ability to move better.
El-Chammas et al. 2011 [33]	RB	---	For AR refer to [33,34]	Cholesterol, anti-IgA, anti-IgM, anti-PC	The actual role of gluten per se is unknown ↓in the activity and levels of antibodies to β-lactoglobulin, and gliadin
Lerner et al. 2017 [34]	RB	---	For AR refer to [33,34]	Immunogenicity, pathogenicity, intestinal permeability, microbiome, oxidative stress, epigenetic programming, cellular metabolism, cognitive function	There is no clear indication for eliminating gluten in non-coeliac patients. Human clinical studies are needed
Badsha 2018 [35]	RB	---	MD, flavonoids/isoflavones, gluten, eD, DV, ED, polyunsaturated fatty acids, probiotics, alcohol, vitamins, supplements, antioxidants, A, obesity	---	Limited evidence to link gluten-free diets to benefits in RA
Vijayalakshmi et al. 2018 [36]	RB	---	DAO, DAI, DVSG, DVSG, eD, MD, ED, DE, supplementation	Refer to [33]	DVSG may be beneficial for some patients as indicated by Hafström et al. 2001 [16].
Lerner et al. 2019 [37]	RB	---	Refer to [30,33,34,36]	Refer to [30,33,34,36]	As the gluten-free diet has not been investigated in isolation for this nutrient it is premature to conclude that it has beneficial effects on RA.
Bruzzese et al. 2020 [38]	*n* = 4 10 days-10 months EC	DSG in patients with RA (no control)	Not described	DAS28	Case 1: improvement observed at 1 month, 5 and 10 months Case 2: improvement observed after one month, but patient stops the diet after three months. Case 3: improvement observed after one month, but patient stops the diet. Case 4: after 10 days, improvement is observed.
Rondanelli et al. 2020 [39]	RB	---	They refer to studies analyzing the intake of potatoes, free sugars, and the recommendation of the DGA as healthy foods (fruits, vegetables, whole grains, nuts, fatty acids).	---	For carbohydrates, three servings per day of whole grains, preferably gluten-free, are recommended. Exclusion of sugars from the diet.
Guagnano et al. 2021 [40]	*n* = 40 3 months RCT	ED Vs. MD	ED = Mediterranean type excluding meat, gluten and dairy products Standard Mediterranean diet, without red meat	DAS28, HAQ, VAS, glucose, insulin, cholesterol, ESR, CRP, transaminases, total protein, albumin, transferrin, cytokines, adipokines, BMI, blood pressure, BIA	Results limited to 28 patients because 12 drop out before 3 months. Differences in VAS and HAQ for ED group pre-post diet; no differences in DAS28. No inter-group comparison. In anthropometric measurements, intergroup differences were only observed for muscle mass, water and systolic pressure. For analytics there are no inter-group differences. For cytokines and adipokines, without intergroup comparison, there is variation only for leptin in each of the two groups.
Jiang et al. 2021 [41]	RB	---	MD, DV, DSG, A,	---	Refer to results of [16,27].

* Where appropriate; A = fasting; ACR = American College of Rheumatology; RA = rheumatoid arthritis; BIA = bioimpedance analysis; CCT = controlled clinical trials; DAI = anti-inflammatory diet; DAO = anti-oxidant diet; DAS28 = disease activity in 28 joints; DE = balanced diet; DGA = *Dietary Guidelines for Americans;* DGA = Dietary *Guidelines for Americans*; BMD = ordinary mixed diet; GFD = gluten-free diet; VD = vegan/vegetarian diet; VVGD = vegan/vegetarian gluten-free diet; CE = case study; RCT = randomised single-blind clinical trial; ECP = eosinophil protein; eD = elemental diet; ED = elimination diet; EP = prevalence study; ECP = eosinophil protein; BMI = body mass index; ESR = erythrocyte sedimentation rate; Hb = haemoglobin; HDL = high-density lipoprotein; IGF-1 = insulin-like growth factor; BMI = body mass index; LDL = low-density lipoprotein; MD = Mediterranean diet; HAQ = *health assessment questionnaire*; IGF-1 = insulin-like growth factor; MD = Mediterranean diet; MPO = myeloperoxides (neutrophils); NO = nitric oxide; OxLDL = oxidised LDL; PC = phosphorylcholine; CRP = C-reactive protein; TF = triceps skinfold thickness; RB = literature review; RCT = randomised controlled trial; RS = systematic review; TG = triglycerides; VAS = *Visual Analogue Scale;* VAS = *Visual Analogue Scale; TF = triceps* skinfold thickness.

## Data Availability

No applicable.

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
