# Peer review of "Evaluation of Gluten Exclusion for the Improvement of Rheumatoid Arthritis in Adults"

_nutrients, 2022, doi:10.3390/nu14245396_

Round 1

Reviewer 1 Report

The article “Evaluation of gluten exclusion for the improvement of rheumatoid arthritis in adults” Provides an assessment that no scientific evidence was found to recommend the exclusion of gluten in patients with rheumatoid arthritis. However, I have some suggestions for this mannuscript.

1.     The source of the dose design needs to be clearly stated.

2.     The author needs to add the basis for the number of included cases.

3.     In the section “Discussion”, the author needs to discuss the results with more references.

4.     The author should strengthen the language editing.

5.     Frontier analysis is not thorough enough.

Author Response

Dear reviewer:
Thank you for reading our document and telling us the appropriate improvements. We have improved the following: Revised language and wording with new author, revised points of improvement in the manuscript. Delivery of the final manuscript. Thank you for your attention. Below we clarify the following points:

1. The source of the dose design should be clearly stated.

The database for the study design was EMBASE. Pubmed, Scopus and Epistemonikos were also consulted. Data extraction for the design was only from EMBASE. Thank you for the suggestion.

2. The author needs to add the basis for the number of cases included.

The initial articles were 288, but only 16 were included in the analysis and discussion.

3. In the "Discussion" section, the author needs to discuss the results with more references.

Thank you for the suggestion. We have revised and improved the English for better understanding.

4. The author needs to strengthen the editing language.

We have added a new author who has improved the English and translated it into British English. A new manuscript is attached.

Reviewer 2 Report

1. Arthritis and gluten (1.2.). 

1.2 appears twice in the manuscript.

2. “Table 1. Eligibility criteria”?  Is it table title or table content?  Please redefine the format of the three lines table.

3. Please unify the font format of the first five lines in Table 2, bold or not?

4. For the information part in Table 2, put a word in the table on one line.

5. The format of paragraphs in the results should be consistent.

6. I suggest that the author can sort out and classify the results according to the research, so as to clearly and intuitively understand the author's views.

7. I strongly recommend checking the format of the table in the final manuscript. And the English needs to be improved on stylistics, grammar level and correct typing mistakes.

8.101 lines of the word Brussels should be lowercase initialsï¼›

9.131 lines of confusing punctuationï¼›

10. Line 149 is missing the separator between Ca and Seï¼›

11. In the results, the format of the fifth paragraph needs to be consistent with the other paragraphs.

12. The whole manuscript should be improved

Author Response

Dear Reviewer:
Thank you for reading our document and telling us the appropriate improvements. We have improved the following: Language and writing revision with new author, revision of improvement points in the manuscript. Final manuscript submission. Thank you for your attention.

Corrections added:

  1. Arthritis and gluten (1.2.). 

1.2 appears twice in the manuscript.

                  Corrected. Numbering is now 1.3

  1. “Table 1. Eligibility criteria”?  Is it table title or table content?  Please redefine the format of the three lines table.

                  Yes; that was the title. Format redefined

  1. Please unify the font format of the first five lines in Table 2, bold or not?

                  Done

  1. For the information part in Table 2, put a word in the table on one line.

                  Format of the table completely reviewed and changed

  1. The format of paragraphs in the results should be consistent.

                  Reviewed and changed

  1. I suggest that the author can sort out and classify the results according to the research, so as to clearly and intuitively understand the author's views.

                  There are different criteria to organise the results when doing a review. For us, the criteria used, date of publication, seemed the best so as to understand how the research evolved in time and how many recent publications mentioned previous ones that were not really conclusive.

  1. I strongly recommend checking the format of the table in the final manuscript. And the English needs to be improved on stylistics, grammar level and correct typing mistakes.

                  Done

8.101 lines of the word Brussels should be lowercase initialsï¼›

                  Done

9.131 lines of confusing punctuationï¼›

                  Done

  1. Line 149 is missing the separator between Ca and Seï¼›

                  Done

  1. In the results, the format of the fifth paragraph needs to be consistent with the other paragraphs.

                  Done

  1. The whole manuscript should be improved

                  Done
